# Progression after First-Line Cyclin-Dependent Kinase 4/6 Inhibitor Treatment: Analysis of Molecular Mechanisms and Clinical Data

**DOI:** 10.3390/ijms241914427

**Published:** 2023-09-22

**Authors:** Federica Villa, Alessandra Crippa, Davide Pelizzoni, Alessandra Ardizzoia, Giulia Scartabellati, Cristina Corbetta, Eleonora Cipriani, Marialuisa Lavitrano, Antonio Ardizzoia

**Affiliations:** 1Medical Oncology, Oncology Department ASST Lecco, 23900 Lecco, Italy; a.crippa@asst-lecco.it (A.C.); d.pelizzoni@asst-lecco.it (D.P.); c.corbetta@asst-lecco.it (C.C.); e.cipriani@asst-lecco.it (E.C.); a.ardizzoia@asst-lecco.it (A.A.); 2School of Medicine and Surgery, University of Milano-Bicocca, 20126 Milano, Italy; a.ardizzoia@campus.unimib.it (A.A.); marialuisa.lavitrano@unimib.it (M.L.); 3Medical Oncology, Fondazione IRCCS San Gerardo dei Tintori, 20900 Monza, Italy; g.scartabellati001@unibs.it; 4Department of Medical and Surgical Specialties, Medical Oncology, University of Brescia, 25121 Brescia, Italy

**Keywords:** metastatic breast cancer, CDK 4/6 inhibitors, cell-cycle-specific and non-specific resistance

## Abstract

Cyclin-dependent kinase 4/6 inhibitors (CDK4/6iss) are widely used in first-line metastatic breast cancer. For patients with progression under CDK4/6is, there is currently no standard treatment recommended at the category 1 level in international guidelines. The purpose of this article is to review the cellular mechanisms underlying the resistance to CDK4/6is, as well as treatment strategies and the clinical data about the efficacy of subsequent treatments after CDK4/6is-based therapy. In the first part, this review mainly discusses cell-cycle-specific and cell-cycle-non-specific resistance to CDK4/6is, with a focus on early and late progression. In the second part, this review analyzes potential therapeutic approaches and the available clinical data on them: switching to other CDK4/6is, to another single hormonal therapy, to other target therapies (PI3K, mTOR and AKT) and to chemotherapy.

## 1. Introduction

Breast cancer (BC) is the most commonly diagnosed cancer in women worldwide [1]. An estimated 5–10% of patients are diagnosed at the metastatic stage of the disease [2]. Some studies have found that approximately 20–30% of patients with early BC may recur with metastatic BC (mBC), although the data are limited [3,4]. Up to 70% of mBC patients have luminal BC, which is defined by estrogen receptor (ER) positive (+) and human epidermal growth factor 2 (HER2) negative (−) expression [5], with a median overall survival (OS) of as long as 57 months [6]. Endocrine therapy (ET) is a preferred option for the treatment of these patients [7].

Cyclin-dependent kinase (CDK4/6), as well as its target protein, cyclin D1, is involved in cell cycle regulation and has been implicated in the pathogenesis of BC and the potential development of endocrine resistance [8]. Several large, randomized trials have demonstrated substantial clinical benefit from the use of CDK4/6 inhibitors (CDK4/6is) (palbociclib, ribociclib and abemaciclib) in the first-line setting for metastatic hormone receptor HR+/HER2− disease, with a substantial improvement in progression-free survival (PFS) [9,10,11,12]. OS benefit has also been demonstrated for ribociclib plus letrozole/fulvestrant, with a median OS of greater than 5 years in patients receiving CDK4/6is [13]. Abemaciclib, in the MONARCH 3 study at the second ad interim analysis, showed a numerical improvement in the OS (an increase in the median OS by >12 months with the addition of abemaciclib to aromatase inhibitor (AI)) but without reaching statistical significance [14]. A follow-up is ongoing for the final OS analysis. Palbociclib failed to demonstrate an overall survival benefit in comparison to ET alone [15]. Whether this is related to the differences in drug efficacy or in patient populations or due to patients being lost to survival follow-up remains unclear. Most patients, however, develop resistance to CDK4/6is in about 2–3 years. For patients with progression under CDK4/6is, there is currently no standard treatment recommended at the category 1 level in international guidelines. Reasonable options include switching to another ET monotherapy, cytotoxic chemotherapy and ET with everolimus (a mammalian target of rapamycin (mTOR) inhibitor), talazoparib or olaparib (poly (ADP-ribose) polymerase (PARP) inhibitors) for patients with germline BRCA mutations, or alpelisib (a phosphoinositide 3-kinase (PI3K) inhibitor) for patients with somatic PIK3CA mutations. Whether or not the CKD4/6i treatment should be continued after the initial disease progression is currently unknown. The median OS of about 5 years achieved with CDK4/6is in the first-line setting with a median PFS of 25–28 months suggests that there is limited efficacy in subsequent antineoplastic treatments. The data on subsequent treatment options and their efficacy are limited. Moreover, most of these data were obtained prior to the widespread use of a CDK4/6is. Understanding the molecular mechanisms of CDK4/6is resistance is crucial to develop prospective trials.

This article will review the cellular mechanisms underlying the resistance to CDK4/6is, as well as treatment strategies and clinical data about the efficacy of subsequent treatments after CDK4/6is-based therapy.

## 2. CDK4/6 Inhibitors’ Resistance Mechanisms

Cyclin-dependent kinase (CDK) 4 and its functional homolog CDK6 are two structurally related kinases with biochemical and biological similarities. Despite having few differences in some of their activities, these enzymes are constantly expressed throughout the cell cycle and, with their partners, D-cyclins, are fundamental for integrating mitogenic and antimitogenic extracellular signals, among which stimulating factors, cytokines, cell–cell contacts and other factors are included, representing a boundary between the environment and the cell cycle machinery [16,17]. The cyclin D-CDK4/6 complex is a driving force that controls the transition from the G1 to the S phases. Also, the INK4 (the cyclin D-CDK4/6 inhibitor molecule) retinoblastoma protein (pRb) pathway regulates cellular proliferation by controlling the G1 to the S cell cycle checkpoint. The dysregulation of this pathway is frequently observed in cancer and contributes to cell cycle progression and persistent growth. CDK4/6 mediates the transition from the G1 phase to the S phase by associating with D-type cyclins and regulating the phosphorylation state of pRb [18]. Unphosphorylated pRb binds and represses the functions of the E2 family (E2F) transcription factors; upon phosphorylation, pRb dissociates from the E2F transcription factors, freeing them to be able to participate in DNA replication and cell division [19].

For these reasons, CDK4/6 has become a treatment target. However, it is common to develop resistance and for the disease to progress during CDK 4/6i treatment. Here, we summarized the various cell-cycle-specific and cell-cycle-non-specific resistance mechanisms depending on the biological determinant of resistance, and in mechanisms conditioning early or late progression, based on the timing of the onset of resistance.

### 2.1. Cell-Cycle-Specific Mechanisms

Multiple factors involved in the regulation of the cell cycle are associated with a resistance to CDK4/6is, the loss of drug target genes, and the overexpression of other genes involved in the progression of the cell cycle (Figure 1).

#### 2.1.1. pRb Loss or Mutations

pRb is a tumor suppressor protein coded by the RB transcriptional corepressor 1 (RB1) that plays a pivotal role as a cell cycle checkpoint factor. In fact, it is involved in the control of the CDK4/6 pathway, one of the main targets of the CDK4/6is drugs and whose loss, most frequently caused by inactivating mutations of RB1, is the main reason for the resistance to CDK4/6is [20,21]. In this case, in spite of this loss, the cell cycle progresses through other molecular pathways, such as the E2F and the cyclin E-CDK2 axis, thus bypassing its dependence on CDK4/6 and causing resistance to CDK4/6is [22]. It has been suggested that administration of cyclin E-CDK2 inhibitors with CDK4/6is may be a valid solution to overcoming resistance [23].

#### 2.1.2. p16 Amplification

p16 is a tumor suppressor protein belonging to the INK4 family involved in cell cycle regulation if pRb is functional. It is an inhibitor of CDK4, and it has been reported to be an accurate biomarker of pRb loss in different tumors [24,25]. p16 amplification is frequently found in BC, and this leads to lower levels of CDK4, thus representing the loss of a target of CDK4/6is. Moreover, p16 amplification is frequently caused by the loss of pRb, thus leading again to resistance to CDK4/6is ex vivo [26]. Palafox et al. suggested that high p16 protein levels and heterozygous RB1 loss-of-function mutations could be predictive in order to identify the CDK4/6is resistance in BC. Moreover, they demonstrated that targeting the p16-CDK4/6 interaction sensitizes p16-overexpressing tumor cells to CDK4/6is [27].

#### 2.1.3. CDK2 Amplification

Cyclin E is encoded by the CCNE1 gene and its association with CDK2 is involved in cell cycle progression from phase G1 to S. Its main role is the pRb phosphorylation, resulting in complete release of E2F [28]. It has been widely reported that CCNE1 overexpression is responsible for the resistance to CDK4/6is. In fact, cells losing their dependence on CDK4/6 use bypass mechanisms, such as CDK2 amplification, to keep up with cell cycle progression [29,30]. Moreover, there is a study that reports that the CDK2 upregulation in ER+ breast cancer is also controlled by TROJAN. TROJAN is a noncoding RNA that can bind to NKRF (NF-Kappa-B-repressing factor) and inhibit its interaction with RELA. This binding leads to CDK2 overexpression and, as aforementioned, leads to CDK4/6is resistance [31].

#### 2.1.4. E2F Amplification

As a pRb transcription factor, E2F plays an important role in cell cycle regulation. As aforementioned, D-CDK4/6 phosphorylates the pRb leading to E2F release with concurrent transcription of proteins (such as cyclin E) necessary for cell cycle progression. At the same time, cyclin E transcription leads to cyclin E-CDK2 formation, which further phosphorylates pRb. For these reasons, the E2F amplification leads cells to evade CDK4/6 cell cycle regulation thus, driving resistance to their inhibitors [18,20,21,28].

#### 2.1.5. CDK7 Overexpression

CDK7 regulates the cell cycle by activating the CDK-activating kinase (CAK) and participates in G1 and G2 phases [32]. Martin et al. reported that its overexpression is involved in CDK4/6is resistance [33]. However, details of its involvement in CDK4/6is resistance are not clear, and further studies are needed to better understand its role.

#### 2.1.6. CDK6 Amplification

The CDK6, together with the CDK4, plays a pivotal role in the cell cycle progression, and its function is mainly kinase-dependent. However, it also upregulates the transcription of p16 in the presence of STAT3 and cyclin D in a kinase-independent way [34]. Moreover, together with c-Jun, CDK6 upregulates the vascular endothelial growth factor A (VEGF-A) which is responsible for cancer progression and drug resistance due to its ability to induce angiogenesis [35]. It was also demonstrated that a particular drug that specifically degrades CDK6 was able to overcome the CDK4/6is resistance [27].

#### 2.1.7. WEE1 Overexpression

WEE1 is a serine/threonine kinase involved in guaranteeing an accurate DNA replication and, in coordination with CDK1, it is responsible for inhibiting DNA-damaged cells from entering mitosis. Most importantly, it plays a key role in the transition from G2 to M phase of the cell cycle. Its role in the CDK4/6is resistance is not clear; however, it was reported that its inhibition in resistant cells could partially restore the CDK4/6 sensitivity [36].

#### 2.1.8. MDM2 Overexpression

Mouse double minute 2 homolog (MDM2) is a negative regulator of p53 which is involved in the activation of p21 (a CDK inhibitor), thus leading to cell cycle arrest [37,38]. MDM2 overexpression leads to interruption of cell senescence and hence CDK4/6is resistance. For this reason, MDM2 inhibitors may be useful in treating patients’ resistant to CDK4/6is, despite further studies being needed [39].

#### 2.1.9. HDACs Activation

Histone deacetylases (HDACs) are responsible for removing the acetyl group from the histones’ ε-N-acetyl lysins, playing a crucial role in gene expression regulation [40]. Moreover, they inhibit p21, which interacts with cyclin D [41]. Recently, it has been reported that the CDK4/6i palbociclib resistance may be conferred by HDAC5 depletion through disruption of palbociclib-induced histone deacetylation and suppression of oncogenic gene expression. Thus, the HDAC5 deficiency in cancer cells could be useful to drive the clinical use of CDK4/6is [42].

#### 2.1.10. FZR1 Loss

Fizzy and the cell division cycle 20-related 1 (FZR1) is a co-activator of the ubiquitin-ligase APC/C that, once activated, can interact with pRb during the cell cycle phase G1 [43]. Ruijtenberg et al. showed that FZR1 is a substrate of cyclin D-CDK4/6 and that, once phosphorylated, it loses its ability to activate APC/C. They further demonstrated that knockdown of both FZR1 and APC/C leads cells to bypass their dependence on cyclin D-CDK4/6 for the progression of the cell cycle [44]. The involvement of the APC/C-FZR1 complex in the downregulation of CDK2/4/6 is particularly interesting because it can also upregulate p27 (a CDK inhibitor) through S-phase kinase-associated protein 2 (SKP2) degradation [45]. For all these reasons, the FZR1 loss is linked to resistance to CDK4/6is.

#### 2.1.11. TK1 Activation

Thymidine kinase 1 (TK1) is an enzyme important for thymidine metabolism during the synthesis of DNA. In resting cells, TK1 activity is low and it increases gradually until it reaches its peak during the S phase. However, it was reported to be continuously overexpressed in patients with different malignancies [46]. It has been demonstrated that TK1 activity is associated with OS and PFS in patients with advanced BC [47]. Moreover, Del Re et al. analyzed exosomal mRNA expression of TK1 and CDK9 in patients with ER+/HER2− mBC enrolled in the ECLIPS biomarker study, showing that higher mRNA expression levels of both of them were linked to palbociclib resistance [48].

#### 2.1.12. miRNAs Expression

In recent years, microRNA (miRNAs) expression in cancer has been widely studied, due to its important role in cancer progression [49]. Among the different miRNAs identified, some were found to be linked to CDK4/6is resistance, such as miR-432-5p, miR-223, and miR-106b [50]. In particular, miR-432-5p was reported to induce resistance through CDK6 overexpression [51]. Moreover, the oncogene c-Myc could reduce the inhibitory effect of miR-29b-3p on CDK6 by downregulating miR-29b-3p, thus inducing BC resistance to palbociclib [52].

#### 2.1.13. S6K1 Amplification

Ribosomal protein S6 kinase beta-1 (S6K1) is a serine/threonine protein kinase that acts downstream of the mTORC1 complex and regulates cell size, protein translation and cell proliferation [53]. Recently, it has been reported that S6K1 amplification is responsible for primary resistance to CDK4/6is, mainly linked to c-Myc overexpression that induces hyperactivation of cyclins/CDKs. The authors also suggested the use of mTOR inhibitors combined with CDK4/6is to overcoming resistance [54].

### 2.2. Cell-Cycle-Non-Specific Mechanisms

Cell-cycle-non-specific mechanisms include the overexpression of factors that are upstream of the cell cycle, such as FGFR and PI3K/AKT/mTOR, resulting in decreased efficacy of CDK4/6is (Figure 2).

#### 2.2.1. FGFR Pathway Activation

The fibroblast growth factor receptor 1 (FGFR1) is a tyrosine kinase involved mostly in cell proliferation, differentiation and survival [55]. Mutations in FGFR1 have been widely reported in different cancers, including BC [56]. Different studies report that FGFR1 and FGFR2 are linked to CDK4/6is resistance. In particular, it was reported that this resistance was caused by the amplification of FGFR1, which led to the activation of PI3K/AKT and RAS/MEK/ERK signaling pathways [57]. In addition, a recent study demonstrated that giving a specific FGFR1 tyrosine kinase-inhibitor to resistant cells could revert the resistance [58]. Finally, it was shown that the FGFR2-activating mutation in ER+ BC could contribute to palbociclib resistance and, when the cells were given a high dose of FGFR inhibitors, they could be completely resensitised to the drug. [59]. These studies suggest a potential therapeutic approach to overcoming such resistance.

#### 2.2.2. PI3K/AKT/mTOR Pathway Activation

Many studies have reported a main role of the PI3K/AKT/mTOR activation in CDK4/6is resistance. In particular, high levels of pyruvate dehydrogenase kinase 1 (PDK1, a protein kinase that acts downstream of PI3K) and activation of the AKT pathway (phospho-S477/T479 AKT) were reported in ribociclib-resistant BC cells. Moreover, inhibition of PDK1 led to higher sensitivity to ribociclib in these cells [60]. Another study revealed that mTORC1/2 may also be involved in CDK4/6is resistance: inhibition of mTOR in ER+ BC results in a decrease in cyclin D1 protein, pRb phosphorylation, and so E2F mediated transcription. In addition, they found that even if cells resistant to CDK4/6is reactivated the CDK-pRb-E2F pathway, they still were sensitive to mTORC1/2 inhibitors [61]. Also, the protein phosphatase and tensin homolog (PTEN) plays a pivotal role in the regulation of the AKT/mTOR pathway. Costa et al. showed that its loss caused a CDK4/6i (ribociclib) resistance by AKT activation. Its loss translated into p27 downregulation which, again, turned into activation of CDK2 and CDK4 [62]. Moreover, knockdown of PTEN in CDK4/6is-sensitive cell lines led to the upregulation of CDK6 and resistance to abemaciclib [63]. These data suggest a wide range of drugs that could be used in combination with CDK4/6is in order to prevent/revert resistance.

#### 2.2.3. MAPK Pathway Activation

The mitogen-activated protein kinase (MAPK) pathway (RAS/RAF/MEK/ERK) is one of the main pathways downstream of FGFR1. It has been reported that selumetinib, a MEK1/2 inhibitor, in association with fulvestrant and palbociclib could inhibit BC proliferation in patients resistant to CDK4/6is [64]. Moreover, it was shown how the overexpression of KRAS, a member of the RAS family, was involved in palbociclib and fulvestrant resistance [65].

#### 2.2.4. Hippo Pathway Inhibition by FAT1

The Hippo pathway is a pathway reported to have a tumor-suppressive role [66]. FAT atypical cadherin 1 (FAT1) is a cadherin that interacts both with the Hippo pathway and β-catenin, and it is reported to act as a tumor suppressor gene [67]. Interestingly, a study of 348 patients treated with CDK4/6is showed loss of FAT1 in patients who became resistant to CDK4/6is. The resistance was probably caused by the fact that FAT1 loss led to Hippo pathway inhibition, which turned into CDK6 overexpression [68]. However, another study reported that more genetic alterations are needed besides FAT1 loss to have CDK6 overexpression [63]. This suggests that more in-depth studies need to be performed to clarify this pathway and to understand whether its targeting could have a therapeutic effect.

#### 2.2.5. Epithelial–Mesenchymal Transition

Epithelial–mesenchymal transition (EMT) consists of the transition of cells from an epithelial to a mesenchymal phenotype. EMT is linked with tissue morphogenesis. However, it has been widely reported to be linked also with tissue invasion, drug resistance and metastasis of cancer [69]. It has been reported that inhibition of the CDK4/6 can induce EMT by activating TGF-β-Smad and PI3K/AKT/mTOR pathways. Once activated, TGF-β can phosphorylate and so activate Smad2 and Smad3. These can create a complex with Smad4 and activate EMT transcription factors [70]. In addition, Smad3 inhibition leads to CDK4/6is resistance, probably because it is no longer able to block E2F from the pRb-E2F complex [71,72]. Moreover, TGF-β can lead to EMT through activation of the PI3K/AKT/mTOR pathway [73]. Recently, it has also been reported that the fibronectin/DHPS/SLC3A2 signaling axis may be involved in the resistance to CDK4/6is. Galler et al. showed that the targeting of this signaling axis improved palbociclib sensitivity in pRb-negative BC cells [74].

#### 2.2.6. Apoptosis Failure

The combination of ET with CDK4/6 inhibition has a predominantly cytostatic effect, thus leading to reduced apoptosis [75]. For this reason, it has been suggested that a combination therapy using CDK4/6is and BCL2 inhibitors could inhibit proliferation and induce apoptosis of cancer cells. Moreover, this approach reduced the proliferation of regulatory T cells in the tumor microenvironment [76].

#### 2.2.7. Stemness Properties

In the last few years, many studies have reported a central role of cancer stem cells (CSCs) in drug resistance. CSCs are a population of cells capable of self-renewal and differentiation potential. Moreover, they show many alterations in different pathways, and for different reasons, and at dates not completely clear, they also play a main role in drug resistance [77]. Interestingly, Wang et al. found that ER+ BC cells treated with palbociclib developed resistance by showing a senescence-like phenotype, which went on to promote stemness. Particularly, PFKFB4 played a major role in this transformation by enhancing glycolysis, and its downregulation resulted in higher palbociclib sensitivity [78]. Thus, even if more studies are needed, targeting CSCs could be a promising therapeutic approach.

## 3. Short- and Long-Term Adaptation to CDK4/6 Inhibitors

CDK4/6 inhibition in BC cells is limited by the inability to induce complete and durable cell-cycle arrest. The mechanisms involved are categorized into two classes: short-term/de novo and long-term adaptation. This distinction is important for the clinic to distinguish the onset of the resistance [79].

### 3.1. Short-Term Adaptation to CDK4/6 Inhibitors

Early adaptation can be mediated by persistent G1-S-phase cyclin expression and CDK2 signaling. Research led by Herrera-Abreu et al. found that early adaptation to CDK4/6 blockade is mediated by the non-canonical cyclin D1/CDK2 complex promoting pRb phosphorylation recovery. In fact, in response to chronic CDK4/6 inhibition, there is a PI3K-dependent upregulation of cyclin D1 along with CDK2-dependent pRb phosphorylation and S-phase entry [30].

A second mechanism that could lead to short-term adaptation to CDK4/6 inhibitors is the promotion of a proinflammatory, senescence-associated secretory phenotype (SASP) in the stroma [80]. These patients have a poor prognosis. Early identification is necessary for cases of de novo or primary resistance, characterized by disease progression within 6 months of initiating treatment [79].

### 3.2. Long-Term Acquisition of CDK4/6 Inhibitors Resistance

On the other side, acquired resistance is a widespread phenomenon, prompting vigorous exploration of various targets, agents, and treatment strategies [79].

Prolonged exposure to CDK4/6is eventually leads to the creation of cell populations that show different resistance mechanisms, like the loss or mutation of pRb, which are the most frequently observed changes in CDK4/6is-resistant cells [78,79]. While this was first observed in preclinical data [30], examples of RB1-inactivating mutations in ctDNA, acquired during the treatment with CDK4/6 inhibitors in patients, have begun to emerge [81,82].

A second known long-term mechanism of resistance is the upregulation of CDK6. Its kinase function is essential for resistance, proven by the correlation between the dose of the kinase inhibitor necessary to cause the pRb loss and the dose that blocks cell proliferation. CDK4 expression, on the other hand, can be reduced in resistant cells since the partner cyclin or other components of the complex influence inhibitor response [83]. The overexpression of CDK2 and CDK4 are amenable to a drug holiday, leading to re-sensitization to palbociclib in vitro and in vivo. Notably, the palbociclib-resistant cells that retain pRb expression are sensitive to abemaciclib, possibly because of increased expression of CDK4 acting as a compensatory mechanism [33]. Other resistance mechanisms found in treatment with CDK4/6is are high levels of cyclins D1 and D2 [60]. At last, CDK6 with c-Jun upregulates VEGF-A, inducing tumor angiogenesis, thus promoting cancer progression and drug resistance [35,84].

In non-cancerous cells, cyclin E-CDK2 (cyclin E1-CDK2 or cyclin E2-CDK2) complexes phosphorylate pRb after a first phosphorylation by cyclin D-CDK4/6 as part of a second phase of signaling. CDK4/6is have multiple effects on the CDK2 action. Without the phosphorylation of the pRb by D-CDK4/6, CDK2 cannot efficiently phosphorylate pRb to release transcription factors like E1 [85]. Different studies have demonstrated an upregulation of cyclin E1, cyclin E2 and CDK2 in CDK4/6is resistance models [30,33,83]. Etemadmoghadam et al. showed how CDK2 inhibitors reduced the growth of cells overexpressing cyclin E1 [29].

In addition to cell-cycle regulation mechanisms, several studies have found that a number of growth factor signaling pathways could determine the insurgence of resistance while in treatment with CDK4/6is. For example, Jansen et al. found that PDK1, the PI3K pathway kinase, was upregulated in cells resistant to ribociclib [60]. Also, in endocrine-resistant cancers, the expansion of FGFR1 activates the PI3K/AKT and RAS/MEK/ERK pathways, both associated with the CDK4/6 resistance [23,64]. PI3K/AKT/mTOR signaling pathway is also involved in the resistance mechanisms: it promotes the stability of the CDK4/6 complex, thus reversing the effects of CDK4/6 inhibition [86].

The detection of acquired resistance allows to explore targets feasible of target therapy, broadening the set of possible therapeutic strategies [79].

## 4. Types of Subsequent Therapies after Randomized Controlled Trial (RCT) and in the Real World

CDK4/6is treatment in first-line therapy was investigated in five RCTs (MONALEESA-2/7, MONARCH-3, PALOMA-1/2): the first subsequent antineoplastic therapy was ET in 65% of cases (min–max 48–83%), chemotherapy in 44% (min–max 32–73%) of cases, CDK4/6 inhibitors up to 38% of cases (on average in 18% of cases) and mTOR inhibitors in 17% of cases (min–max 14–24%) [9,10,11,22,87].

When CDK4/6is was used in second-line therapy (MONARCH-2 and PALOMA-3 RCTs), ET was administered on average in 55% of cases (min–max 37–71%), chemotherapy in 66% of cases (min–max 56–76%), CDK4/6is in 9% of cases (min–max 2–21%) and mTOR inhibitors in 24% of cases (min–max 15–33%) [88,89].

In MONALEESA-3, patients treated with fulvestrant and the CDK4/6/placebo inhibitor in first- and second-line therapy received ET (55%), chemotherapy (43%), CDK4/6 inhibitor (20%) and mTOR inhibitor (30%) [90].

Real-world studies suggest that chemotherapy and ET are the preferred second-line options. In a population-based study, second-line regimens were ET in 38%, chemotherapy in 35.6%, everolimus-based (mTOR inhibitor) in 14.4% and CDKi-based in 9.4% [91]. In a retrospective real-world analysis, cytotoxic chemotherapy was the most commonly chosen second-line therapy (29.7%), followed by endocrine monotherapy (12.4%). The 36.0% of patients continued a CDK4/6is in the second-line treatment setting, either alone or in combination with ET. Other targeted therapies used were everolimus (11.7%), alpelisib (1.9%) and a PARP inhibitor (0.5%) [92].

Little data about the preferred regimen of chemotherapy are available. In a multicenter analysis, taxanes were the most used agents followed by capecitabine and vinorelbine [93].

## 5. Progression after CDK4/6is

Previous case studies reported a rapid, secondary disease progression after the treatment with CDK4/6is: in four patients treated at MD Anderson Cancer Center, a median time-to-progression of 2.35 months (range 1.46–2.8) was reported [94]. In a retrospective monocentric review (abstract presented in ASCO 2021) the 4-month incidence of post CDKi progression or death was 31% [95].

These initial suggestions were not further confirmed.

Progression-free survival 2 (PFS2), time from randomization to second disease progression or death, is recommended by the EMA as a surrogate for OS, and to assess the effect of maintenance therapy or the impact of treatment on the efficacy of a subsequent line of therapy [96].

In the systematic review by Munzone et al., a PFS2 benefit was observed in patients who received CDK4/6is plus ET (pooled hazard ratio (HR) 0.68, 95%, confidence interval (CI) 0.62–0.74). A delay in subsequent time to chemotherapy (TTC) (pooled HR 0.65, 95% CI 0.60–0.71) was reported as well [97].

## 6. Potential Therapeutic Approaches

### 6.1. Continuation of CDK4/6is

The role of CDK4/6is persistance after initial progression was investigated in several retrospective and a few prospective trials. Four single-institution experiences and one multicenter experience showed an interesting clinical benefit of this strategy [98,99,100,101,102]. Several studies explored the use of abemaciclib after initial progression to palbociclib.

The Taussig Cancer Institute followed 30 patients who continued CDK4/6is beyond the first progression. Most patients received palbociclib plus letrozole as initial therapy (67%), followed by palbociclib plus fulvestrant (23%) and palbociclib plus anastrozole (10%). At progression, most patients received palbociclib plus fulvestrant (56.7%) followed by palbociclib plus AI (23.3%), palbociclb plus tamoxifene (13.3%) and abemaciclib with either fulvestrant or letrozole (6.6%). The estimated median PFS for continued CDK4/6is use beyond the first PD was 11.8 months (95% CI, 5.34–13.13 months) [99].

In another larger analysis by Martin et al., 839 patients received a second-line therapy after progression of first-line CDK4/6is treatment, and 308 patients (36%) continued a CDK4/6is therapy [92]. Palbociclib was the most commonly used CKD4/6i in the first-line setting (88.2%), followed by ribociclib (7.2%) and abemaciclib (4.6%). The most interesting data show that 74.4% of patients in the second-line treatment received the same CDK4/6i of the first-line treatment; in particular, 78.2% of patients who received palbociclib in the first-line treatment received the same CDK4/6i compared to 60.9% of patients who started with ribociclib and 45.8% who started with abemaciclib. The PFS and OS of patients who continued CDK4/6is in the second-line treatment were 8.25 months and 35.7 months, respectively, which are significantly improved compared to chemotherapy (HR PFS 0.48, 95% CI 0.43–0.53, *p* < 0.0001; HR OS 0.30, 95% CI 0.26–0.35, *p* < 0.0001).

The prospective trials produced conflicting results (Table 1).

In the MAINTAIN trial (NCT02632045), ET plus ribociclib demonstrated a longer PFS (5.29 months vs. 2.76 months) than ET plus placebo after progression of a first-line treatment with CDK4/6is [103]. Notably, 84% of the patients had previously received palbociclib and more than two-thirds had previously received CDK4/6is for more than 12 months.

However, in the phase II PACE trial (NCT03147287), the median PFS of patients treated with fulvestrant (4.8 months) was the same as in patients treated with both fulvestrant and palbociclib (4.6 months) [104]. The PACE trial randomized patients, after progression on CDK4/6is, to palbociclib plus fulvestrant versus placebo plus fulvestrant versus palbociclib plus fulvestrant plus avelumab. Notably, more than 90% of patients received palbociclib as the initial CDK4/6i and 76% received it for more than 12 months.

In the PALMIRA trial (NCT03809988), presented at ASCO 2023, 198 patients showing a clinical benefit from the first-line with palbociclib plus ET for at least 6 months were randomized at progression to receive another ET different from the first-line treatment plus palbociclib or placebo: median investigator-assessed PFS was 4.2 months (95% CI 3.5–5.8) in the palbociclib arm vs. 3.6 months (95% CI 2.7–4.2) in the ET arm (HR 0.8, 95% CI 0.6–1.1, *p* = 0.206). The authors stated that maintaining palbociclib with a second-line ET beyond progression of prior palbociclib-based therapy did not significantly improve PFS compared with second-line ET alone [105].

The benefits of continuing CDK4/6is after initial progression remain unclear. Whether the results of MAINTAIN, PACE and PALMIRA are related to differences in drug efficacy (palbociclib and ribociclib) or in patient populations (in the PACE and PALMIRA trials, all patients had to be on a CDK4/6i first-line therapy for at least 6 months, while in the MAINTAIN trial there are no limitations) or due to switch to a different CDK4/6i (in PACE and PALMIRA, a rechallenge of palbociclib was performed, while in MAINTAIN, a switch to ribociclib was carried out) remains unclear as well (Table 1).

**Table 1 ijms-24-14427-t001:** Summary of the trials focused on continuing CDK4/6is after progression.

Trial	Number of Patients	Population	Prior CDK4/6is	Subsequent ET	Subsequent CDK4/6is	Efficacy	PFS (Months)
MAINTAIN [103]	119	Progression on CDK4/6is + ET	Palbociclib 86.5% Ribociclib 11.7%	Fulvestrant 83.2% Exemestane 16.8%	Ribociclib	RR 20% CBR 43%	ET + Ribociclib 5.29 ET + placebo 2.76
PACE [104]	220	Progression on CDK4/6is + ET after at least 6 months of therapy	Palbociclib 90.9% Ribocliclib 4.5% Abemaciclib 4.1%	Fulvestrant	Palbociclib	RR ET 10.8% RR palbociclib + ET 3.8% RR Triplet 17.9%	ET 4.8 Palbociclib + ET 4.6 Triplet 8.1
PALMIRA [105]	198	CB during palbociclib	Palbociclib 100%	Fulvestrant Letrozole	Palbociclib	6-months CBR 41.9%	ET + Palbociclib 4.9 ET 3.6
BIOPER [106]	33	CB during palbociclib Palbociclib as last treatment	Palbociclib 100%	Fulvestrant 56% Letrozole 28% Others 15.6%	Palbociclib	CBR 34.4%	2.6

Abbreviation: ET = endocrine therapy; CB = clinical benefit; RR = response rate; CBR = clinical benefit rate; PFS = progression-free survival.

The BioPER trial (NCT03184090) evaluated the antitumor activity, the safety and the predictive biomarkers of palbociclib rechallenge in 33 patients with confirmed progressive disease after having achieved a clinical benefit on immediately prior palbociclib plus endocrine therapy regimen. The clinical benefit rate was 34.4%, but the median PFS was modest (2.6 months). The analysis of biomarkers revealed that low Rb score, high cyclin E1 score and ESR1 mutations were associated with worse outcomes than palbociclib rechallenge [106].

In the future, new prospective clinical trials and more data could clarify if continuing CDK4/6is after progression is a valid option to overcoming resistance, particularly the following:

PostMONARCH (NCT05169567) is a randomized, double-blind, placebo-controlled, phase III trial that compares the efficacy of abemaciclib plus fulvestrant to placebo plus fulvestrant in patients with HR+/HER2−, advanced or metastatic BC following progression on CDK4/6is and ET (recruiting);

Serena-6 (NCT04964934) is a phase III, double-blind, randomized trial that could assess switching to AZD9833 (a next-generation oral SERD) plus CDK4/6is (palbociclib or abemaciclib) vs. continuing AI (letrozole or anastrozole) plus CDK4/6is in HR+/HER2− metastatic BC patients with detectable ESR1 mutation without disease progression during first-line treatment with IA plus CDK4/6is (recruiting).

### 6.2. Endocrine Therapy

In some cases, de-escalation to only ET after progression of CDK4/6is is a possible strategy; as shown by Karacin C. et al., PFS from patients receiving ET after progression on CDK4/6is was not different compared to those receiving chemotherapy [107]. ET could be fulvestrant, AI or tamoxifen.

Fulvestrant is more effective than AI, as demonstrated in the FALCON trial (NCT01602380); PFS was significantly longer in the fulvestrant group than in the anastrozole group (HR 0.797, 95% CI 0.637–0.999, *p* = 0.0486), and mPFS was 16.6 months (95% CI 13.83–20.99) in the fulvestrant group versus 13.8 months (11.99–16.59) in the anastrozole group [108].

However, in these trials, fulvestrant was given as a first-line therapy. Nevertheless, in the second-line treatment after progression of CDK4/6is, the PFS of fulvestrant has been demonstrated to be much lower (EFECT trial, NCT00065325) [109].

In the phase III EMERALD trial (NCT03778931), 477 patients pretreated with a CDK4/6is and ≤1 chemotherapy were randomized to elacestrant, a new oral selective ER degrader (SERD), or standard of care (fulvestrant). Elacestrant had a superior PFS rate compared to fulvestrant, and the HR improved in the subgroup of patients with ESR1-mutated tumors. The mPFS in the intention-to-treat (ITT) population on elacestrant was 2.8 months, compared to 1.9 months for fulvestrant monotherapy [110]. In this study, patients with ESR1 mutation had a higher benefit with a median PFS of 3.8 months with elacestrant and 6- and 12-month PFS rates of 41% and 27%. Particularly, patients who had remained on a prior CDK4/6 inhibitor for ≥12 months showed a PFS benefit of 6 months (median 8.6 vs. 1.91 months, hazard ratio 0.41, 0.26–0.63), indicating that this might be the ideal setting for elacestrant [111].

In a phase II SERENA-2 trial (NCT03616587), another SERD, camizestrant, demonstrated an advantage in PFS compared to fulvestrant (7.7 months vs. 3.7 months). Patients were eligible if they had received ≥1 line of ET and no more than 1 prior chemotherapy regimen. A total of 50% of patients had a prior CDK4/6i, 42% of patients had prior SERM, and 37% of patients had an ESR1 mutation [112].

These data suggested that the use of fulvestrant after progression on the first-line treatment of CDK4/6is could be exploited in the case of limited progression because of low PFS. A more viable alternative to ET is the use of SERD, as it is more effective than fulvestrant.

### 6.3. Switch to Combined Target Therapies

As mentioned above, many studies have reported a main role of PI3K/AKT/mTOR activation in the CDK4/6is resistance: targeting these specific pathways could be another option to overcoming the resistance (Table 2).

#### 6.3.1. ET Combined with PI3K Inhibitors

Approximately 40% of patients with HR+/HER2− advanced BC have PIK3CA (phosphatidylinositol-4,5-bisphosphate 3-kinase catalytic subunit α) mutations. These mutations can promote endocrine resistance through the activation of the PI3K pathway, which is associated with a poor prognosis [113].

In the SOLAR-1 study (NCT02437318), the combination of fulvestrant and alpelisib, an α-selective PI3K inhibitor, was compared with fulvestrant and placebo for patients with HR+/HER2− advanced BC with PIK3CA mutations who progressed during or after the treatment with AI. The PFS time of alpelisib plus fulvestrant group was prolonged by 5.3 months (11 vs. 5.7 months; *p* = 0.00065) and the mOS time was prolonged by 7.9 months (39.3 vs. 31.4 months) [114]. According to these results, alpelisib has been approved by the FDA for marketing. However, <10% of the patients enrolled in the SOLAR-1 study received CDK4/6is treatment, so the drug was approved by EMA only with the specific indication “after disease progression following endocrine therapy as monotherapy” [115].

The BYLieve study (NCT03056755) investigated the question of whether alpelisib plays the same role in patients who progress after treatment with CDK4/6is [116]. Patients with PIK3CA mutations who progressed after first-line treatment were enrolled in two different cohorts: cohort A for patients who received AI combined with CDK4/6is as first-line treatment and received alpelisib combined with fulvestrant as second-line treatment; cohort B for patients who were treated with fulvestrant combined with CDK4/6is as first-line treatment and received alpelisib combined with letrozole as second-line treatment. Updated data results were presented at the ASCO 2022 meeting: mPFS time was reported as 8.2 months in cohort A and 5.6 months in cohort B [117], showing a potential clinical benefit of alpelisib plus endocrine therapy also after a CDK4/6is. Some issues remain about the toxicity profile of the drug: in this trial, about 26% of patients had grade >3 side effects including hyperglycemia, rash and diarrhea.

Other PI3K inhibitors (such as inavolisib, LOXO-783 [118] and RLY-2608 [119]) are currently being studied in this setting in order to maintain the activity of this class of drugs while reducing off-target toxicities.

#### 6.3.2. ET Combined with mTOR Inhibitors

Everolimus is the mTOR inhibitor most commonly used in BC. Its effect has been confirmed in the BOLERO-2 study (NCT00863655): for patients who progressed after NSAI treatment, the everolimus plus exemestane (SAI) group had significantly longer mPFS and mOS than the placebo plus exemestane group (mPFS: 6.93 vs. 2.83 months; HR, 0.43; *p* < 0.0001; mOS: 31.0 vs. 26.6 months; HR, 0.89; *p* = 0.14) [120].

In real-world studies, the combination of everolimus plus exemestane was largely used as second-line therapy after CDK4/6is. One study from the US analyzed 525 patients who received systemic therapy after a CDKi-based line: the second-line regimen (*n*  =  208) after CDK4/6is was everolimus-based in 14.4% of patients [91]. Another recent study analyzed 609 patients and calculated the PFS of subsequent treatments (chemotherapy, n:434 or ET, *n*:175) after CDKi. Patients were evaluated as three groups: those who received CDKi in first-line therapy (group A, *n*: 202), second-line therapy (group B, *n*: 153) and  ≥third-line therapy (group C, *n*: 254). PFS was compared according to the use of ET and chemotherapy. In addition, patients who received ET after CDKi were compared as those who received everolimus-based combination therapy versus those who received ET monotherapy: the median PFS in groups A, B, and C was 11.0 vs. 5.9 (*p*  =  0.047), 6.7 vs. 5.0 (*p*  =  0.164), 6.7 vs. 3.9 (*p*  =  0.763) months, suggesting a role for everolimus in achieving better PFS [107].

At the ASCO 2019 meeting, the TRINITI-1 clinical study (NCT02732119) reported enrolled patients who progressed after treatment with CDK4/6is [121]. After receiving exemestane, everolimus and ribociclib combination therapy, the clinical benefit rate at 24 weeks reached 41%, which exceeds the pre-defined primary endpoint threshold (>10%). The mPFS time of the overall population reached 5.7 months [122].

An ongoing phase III study (evERA, NCT05306340) is currently evaluating the combination of the oral SERD giredestrant and everolimus compared to everolimus and exemestane in patients who have previously progressed onto CDK4/6is [123].

The combination of CDK4/6is with PI3K or mTOR inhibitors is currently being tested in clinical trials because synergistic activity for these drugs has been observed [124], but the studies evaluating these combinations showed severe toxicity. The trial with the combination of palbociclib plus everolimus plus exemestane was limited by frequent high-grade mucositis and neutropenia and also showed limited efficacy [125]. Similar toxicity concerns were observed in another study with the combination ribociclib plus everolimus plus exemestane [126]. Other ongoing phase I trials are investigating the safety and efficacy of combinations including abemaciclib and xentuzumab (an IGF-neutralizing antibody) [127], gedatolisib (an mTOR/PI3K inhibitor) in combination with palbociclib and either letrozole or fulvestrant [128], fulvestrant, palbociclib and erdafitinib (a newer generation FGFR inhibitor) [129].

#### 6.3.3. ET Combined with Other Targeted Drugs

AKT acts as a bridge connecting the PI3K and mTOR signaling pathways. The phase Ib TAKTIC study (NCT03959891) evaluated the efficacy of the AKT-1 inhibitor ipatasertib plus endocrine therapy plus or not palbociclib in patients with HR+/HER2− advanced BC. The results showed that in some patients (8/12) who had failed previous CDK4/6is treatment, the combination of AKT inhibitor plus palbociclib plus endocrine therapy achieved good clinical effects and was well tolerated [130]. More data remain to be seen.

In a recent phase 3, randomized, double-blind trial (CAPItello-291, NCT04305496), 708 patients with HR+/HER2− advanced BC, who had had a relapse or disease progression during or after treatment with an IA, with or without previous CDK4/6is therapy, were randomly assigned in a 1:1 ratio to receive capivasertib (AKT inhibitor) plus fulvestrant or placebo plus fulvestrant. The PFS was 7.2 months in the capivasertib–fulvestrant group as compared with 3.6 months in the placebo–fulvestrant group [131], suggesting an interesting clinical activity of this combination.

HDAC inhibitors are another target therapy of interest. In the ACE study (NCT02482753), the HDAC inhibitor tucidinostat (formerly known as chidamide) combined with exemestane showed longer mPFS time compared with placebo and exemestane (7.4 vs. 3.8 months; *p* = 0.033) [132]. The ENCORE 301 (NCT00676663), a phase II randomized, placebo-controlled study, demonstrated a significant improvement in PFS and OS with the addition of entinostat to exemestane in patients with HR+ advanced BC with disease progression after prior non-steroidal AI. The subsequent phase III E2112 registration trial (NCT02115282) did not confirm the results, with an mPFS of 3.3 months in the entinostat-exemestane group versus 3.1 months in the placebo–exemestane group; the mOS was 23.4 months (entinostat–exemestane) versus 21.7 months (placebo–exemestane); the objective response rate was 5.8% (entinostat–exemestane) and 5.6% (placebo–exemestane). In the E2112 study, 35% of the patients were previously treated with CDK4/6is [133]. Probably, this class of drugs is not so efficient in overcoming CDK4/6is resistance.

Another interesting target is the B-cell lymphoma-2 (BCL2) gene, an estrogen-responsive gene coding for an antiapoptotic protein overexpressed in approximately 80% of patients with HR+ BC [134]. Venetoclax, a potent and selective BCL2 inhibitor, combined with tamoxifen had a tolerable safety profile and significant activity in patients with ER+/HER2− advanced BC that overexpresses BCL2 in a phase I clinical trial [135]. The VERONICA trial (NCT03584009) evaluated venetoclax plus fulvestrant vs. fulvestrant in women with locally advanced ER+/HER2− or mBC, who received ≤2 prior lines of ET and no prior chemotherapy in the locally advanced or mBC setting and experienced disease recurrence/progression during/after CDK4/6is therapy. Preliminary results from the primary analysis did not show a significant difference in PFS time between the combination and monotherapy groups (2.69 vs. 1.94 months; *p* = 0.7853) [136]. An ongoing phase I clinical trial, PALVEN (NCT03900884) [137] is evaluating the safety and efficacy of AI plus CDK4/6is plus venetoclax triple therapy as a first-line treatment for patients with ER+/HER2− advanced BC with BCL2 overexpression, and we are waiting for the results.

#### 6.3.4. CDK4/6 Inhibitors plus Other Targeted Drugs

The combination of CDK4/6is with immune checkpoint inhibitors is under investigation because of the demonstrated activity of CDK4/6is not only in inducing the tumor cell cycle arrest, but also in promoting anti-tumor immunity [138]. CDK4/6is have effects both on tumor cells and on regulatory T cells: they markedly suppress the proliferation of regulatory T cells and indirectly stimulate the production of type III interferons and hence enhance tumor antigen presentation. These events promote cytotoxic T-cell-mediated clearance of tumor cells, which is further enhanced by the addition of immune checkpoint blockade.

The PACE trial, already discussed above, had a third arm with avelumab, fulvestrant and palbociclib. This triplet regimen increased PFS and OS compared with the other two groups, although neither difference achieved statistical significance. The median PFS was 8.1 months (HR 0.75 vs. fulvestrant, 95% CI 0.47–1.20), and the median OS was 42.5 months (HR 0.68 vs. fulvestrant, 95% CI 0.4–1.15). No major toxicity signals were identified. The comparison of a triplet arm versus fulvestrant was a secondary endpoint. The numerical superiority of PFS and OS with avelumab deserves further study [93,104].

An ongoing phase IB clinical trial, JPCE (NCT02779751), is evaluating the efficacy and safety of abemaciclib in combination with pembrolizumab (a PD-1 inhibitor) in the first-line treatment of HR+/HER2− mBC.

**Table 2 ijms-24-14427-t002:** Summary of the perspective trials focused on target therapies (TT).

Target Therapy	Trial	Number of Patients	Population	Prior CDK4/6is	Subsequent TT	Efficacy	PFS (Months)
PI3K inhibitor	SOLAR-1 [114]	572	Progression on AI. Stratification by prior treatment with CDK4/6is	Any	Fulvestrant + Alpelisib	NA	7.3
	BYLieve [116]	127	Progression on no more than two previous anticancer therapies and no more than one previous chemotherapy regimen. Confirmed PIK3CA mutation	a. CDK4/6i + AI b. CDK4/6i + fulvestrant	a. Fulvestrant + Alpelisib b. Letrozole + Alpelisib	PR 17% SD 45%	a. 8.2 b. 5.6
MTOR inhibitor	TRINITI-1 [122]	104	Progression on a CDK4/6i after ≥4 months of therapy as the last prior treatment regimen	Any	Exemestane + Everolimus + Ribociclib	CBR 41%	5.7
AKT inhibitor	CAPItello-291 [131]	708	Progression during or after treatment with an IA, with or without previous CDK4/6is	Any (69.1% of pts)	a. Capivasertib + fulvestrant b. Placebo + fulvestrant	a. ORR 22.9% b. ORR 12.2%	a. 7.2 b. 3.6
HDAC inhibitor	ACE [132]	365	Progression after at least one endocrine therapy	Palbociclib (>1% pts)	a. Tucidinostat + exemestane b. Placebo +exemestane	a. PR 18% SD 56% b. PR 9% SD 54%	a. 7.4 b. 3.8
	E2112 phase III [133]	608	Progression on AI in the adjuvant or metastatic setting	Any (35% of pts)	a. Etinostat + exemestane b. Placebo + exemestane	a. 5.8% b. 5.6%	a. 3.3 b. 3.1
BCL-2 inhibitors	VERONICA [136]	103	≤2 prior lines of ET and prior chemotherapy in the locally advanced/mBC setting and progression during/after CDK4/6is	Any	a. Venetoclax + fulvestrant b. Fulvestrant	a. CBR 11.8% b. 13.7%	a. 2.69 b. 1.94
Immune checkpoint inhibitors	PACE [104]	200	Prior response to and subsequent progression on CDK4/6is and ET	Palbociclib 90% Ribociclib 4.5% Abemaciclib 4.1% Palbociclib + Ribociclib 1.4%	a. Avelumab + Fulvestrant + Palbociclib b. Fulvestrant + Palbociclib c. Fuvestrant	a. 17.9% b. 13.8% c. 10.8%	a. 8.1 b. 4.6 c. 4.8

Abbreviation: TT = target therapy; ORR = objective response rate; CBR = clinical benefit rate; PR = partial response; SD = stable disease; PFS = progression-free survival.

#### 6.3.5. PARP Inhibitors

Two randomized phase III studies, OlympiAD (NCT02000622) and EMBRACA (NCT01945775), demonstrated the efficacy of PARP inhibitors (olaparib and talazoparib) in germline-BRCA-mutated mBC patients: they both showed an improvement in the overall response rate and PFS (median improvement of 3 months) but failed to demonstrate an improvement in OS [139,140,141]. These trials were designed before the CDK4/6is era, so there are very little data about the efficacy of PARP inhibitors after CDK4/6is. However, for selected patients with the germline BRCA mutation, PARP inhibitors could be a reasonable alternative to chemotherapy beyond CDK4/6is in the endocrine refractory setting [142].

### 6.4. Switch to Chemotherapy

Chemotherapy is also a good option for patients with HR+/HER2− advanced BC who are resistant to ET plus CDK4/6is [143]. As mentioned above, considering the five randomized clinical trials with CDK4/6is in a first-line setting (MONALEESA-2/7, MONARCH-3, PALOMA-1/2, NCT01958021, NCT02278120, NCT02246621, NCT00721409 and NCT01740427), patients received chemotherapy as the first subsequent antineoplastic therapy after study discontinuation in an average of 44% (min–max 32–73%) of cases. In MONARCH-2 and PALOMA-3 (NCT02107703 and NCT01942135, respectively) chemotherapy was administered in 66% of cases (min–max 56–76%); patients enrolled in MONALEESA-3 (NCT02422615) treated with fulvestrant and the CDK4/6/placebo inhibitor in the first- and second-line settings subsequently received chemotherapy in 43% of cases [97].

Many real-world studies have been published in the past years (Table 3). One still-mentioned study from the US showed that out of 525 patients who progressed after receiving CDK4/6is treatment, more than a third of the patients received subsequent chemotherapy, and the drugs used were capecitabine and taxanes [91]. Another American study analyzed 1210 patients with HR+/HER2− mBC who were treated in a first-line setting with a CDK4/6i from 2015 to 2020; a total of 839 patients received a documented second-line therapy after progression of first-line CDK4/6is treatment. Chemotherapy was chosen for 29.7% of patients, and the use of chemotherapy decreased over time [92]. Eribulin, a tubulin polymerization inhibitor, demonstrated efficacy after CDK4/6is treatment in a Russian study [144], with an mPFS of 10.0 months; the 6-month, 1-year, and 2-year PFSs were 79.5%, 44.8% and 26.5%, respectively.

Trastuzumab–deruxtecan (T-DXd) is an antibody–drug conjugate consisting of monoclonal antibody trastuzumab with a topoisomerase I inhibitor payload. In the DESTINY-Breast04 trial (NCT03734029), a group of HR+ patients with endocrine-resistant disease was randomized to T-DXd versus the physician’s choice of chemotherapy. The percentage of patients who had a prior CDK4/6i was 78% in the investigational arm and 81% in the control arm. In this HR+ group, patients had an mPFS of 10.1 months on T-DXd versus 5.4 months in the control arm (HR 0.5164, 95% CI 0.40–0.64, *p* < 0.0001). Patients also had a statistically significant improvement in mOS of 6.4 months (HR 0.64, 95% CI 0.48–0.86, *p* = 0.0028). In the T-DXd arm, OS was 23.9 months versus 17.5 months in the control arm. Patients with HR+ disease had an overall response rate of 52.6% and a clinical benefit rate (CBR) of 71.2%, establishing T-DXd as a possible new standard of care post-CDK4/6is [145], especially in patients with rapidly progressive disease. In DESTINY-Breast04, all patients received one or two lines of previous chemotherapy. It will be very interesting to analyze the results of the DESTINY-Breast06 trial, which is evaluating the role of trastuzumab deruxtecan before chemotherapy.

An exploratory analysis of CDK4/6is resistance marker signatures, performed in DESTINY-Breast04, was recently reported [146]. The signature is not yet clinically validated and includes analysis of CCND1, CCNE1, CDK6, and FGFR1/2 amplification and RB1, PTEN, RAS, AKT1, ERBB2, and FAT1 mutations [147]. Notably, improved ORR for T-DXd over TPC was observed regardless of CDK4/6is resistant markers. CDK4/6is resistance markers were also assessed in patients with or without prior CDK4/6is therapy and a longer median PFS was observed in the T-DXd arm compared to the TPC arm regardless of the presence of these markers.

Sacituzumab–govitecan (SG) is an antibody–drug conjugate that targets human trophoblast cell surface antigen 2 (TROP-2) and is designed to effectively deliver a chemotherapeutic agent. It initially received FDA approval for patients with triple-negative mBC, but TROP-2 is overexpressed also in other breast subtypes and has been studied in patients with HR+, HER- mBC. In the multicenter phase III TROPiCs-02 study (NCT03901339), patients with HR+/HER2− advanced BC were randomized 1:1 into SG versus physicians’ choice of chemotherapy. Patients were heavily pretreated because they were required to have progressed onto a CDK4/6i and at least two chemotherapy agents, including a taxane. The study demonstrated statistically significant improvement in PFS on SG of 5.5 months versus 4.0 months on physicians’ choice chemotherapy (HR 0.66; 95% CI, 0.53–0.83; *p* = 0.0003). Patients also had an improved overall response rate of 21% versus 14% on SG, as well as an improved CBR of 34% versus 22% [31]. In the second interim analysis, an OS benefit of 3.2 months was seen (HR 0.79, CI 0.65–0.96, *p* = 0.02). Patients on SG had an OS of 14.4 months compared to 11.2 months for patients on the treatment of their physician’s choice [148], suggesting an interesting role for this drug, maybe after trastuzumab deruxtecan, considering its 6.4 months benefit in OS in the DESTINY-Breast04 trial mentioned above (Table 3).

There are ongoing clinical trials evaluating the efficacy of chemotherapy after CDK4/6 inhibition. In particular, there is the TATEN trial (NCT04251169), investigating pembrolizumab plus paclitaxel in HR+/HER2− non-luminal (by PAM50) advanced BC after CDK4/6is progression, and a phase I study (NCT04134884) to test the safety of a combination of ASTX727 with talazoparib in patients with triple-negative BC or HER2− mBC [149,150].

**Table 3 ijms-24-14427-t003:** Summary of the trials focused on chemotherapies (CT).

Trial	Number of Patients	Population	Prior CDK4/6is	Subsequent CT	Efficacy	PFS (Months)
1-US study [91]	525	Progression on CDK4/6is	Any	Capecitabine or Taxanes (35.6%)	NA	NA
2-US study [92]	1210	Progression on first-line CDK4/6is	Palbociclib 88.2% Ribociclib 7.2% Abemaciclib 4.6%	NA	NA	3.71
Russian study [144]	54	Progression on CDK4/6is	Palbociclib 75.9% Ribociclib 22.2% Both 1.9%	Eribulin 61.1% Others 38.9 %	PR 24.4% SD 66.7%	10.0
DESTINY-Breast04 (HR+ cohort) [145]	494	Progression on CT or OT	Any (about 70% of pts)	T-DXd vs. TPC (eribulin, capecitabine, nab-paclitaxel, gemcitabine or paclitaxel)	52.3% T-DXd vs. 16.3% TPC	10.1 T-DXd vs. 5.4 TPC
TROPiCS-02 [148]	543	Progression after CDK4/6is and at least 2 chemotherapeutic agents (including a taxane)	Any (98% of pts)	SG vs. TPC (eribulin, capecitabine, gemcitabine or vinorelbine)	21% SG vs. 14% TPC	5.5 SG vs. 4.0 TPC

Abbreviation: CT = chemotherapy; OT = ormonal therapy; PR = partial response; SD = stable disease; PFS = progression-free survival; TPC = Treatment of Physician’s Choice; T-DXd = trastuzumab deruxtecan; SG = sacituzumab govitecan.

## 7. Conclusions

The widespread use of CDK4/6is in metastatic BC resulted in a question: what to do next? There is currently no standard treatment after CDK4/6is.

Molecular mechanisms underlying CDK4/6is resistance are very complicated; thus, their clarification is crucial in determining the next treatment plan. According to the different resistance mechanisms, targeted drugs that could be chosen after resistance should also be different.

Several factors are currently considered to select the best treatment after CDK4/6is: the presence or absence of driver mutations (ESR1, PIK3CA, germline BRCA1-2), the duration of exposure to CDK4/6is, the burden and sites of metastatic disease, comorbidities, patient preference and the availability of clinical trials.

In ESR1 mutant patients, especially in those with prolonged exposure to prior CDK4/6is, elacestrant seems to be the best choice.

In PI3KCA-mutated tumors, the combination of fulvestrant plus alpelisib should be evaluated, being aware of the side effects. A less toxic alternative in this setting is capivasertib plus fulvestrant as shown in CAPItello-291 study.

For patients with germline BRCA mutation, a PARP inhibitor could be considered particularly as an alternative to chemotherapy. Another option, notably in patients without driver mutations, could be everolimus plus exemestane.

Continuation of CK4/6i beyond progression remains controversial.

Finally, in patients with rapidly progressive disease, chemotherapy is the most reasonable choice: nevertheless, which agent is the best remains unknown. The ongoing DESTINY-breast06 trial will contribute to answering this question.

In third-line treatment, trastuzumab deruxtecan and sacituzumab govitecan are valid options.

Some novel agents (like BCL2 inhbitors, HDAC inhibitors or AKT-1 inhibitors) or novel combinations (like immune checkpoint inhibitors with CDK4/6is) are also promising. For these reasons, patient enrollment in clinical trials should be encouraged.

## Figures and Tables

**Figure 1 ijms-24-14427-f001:**
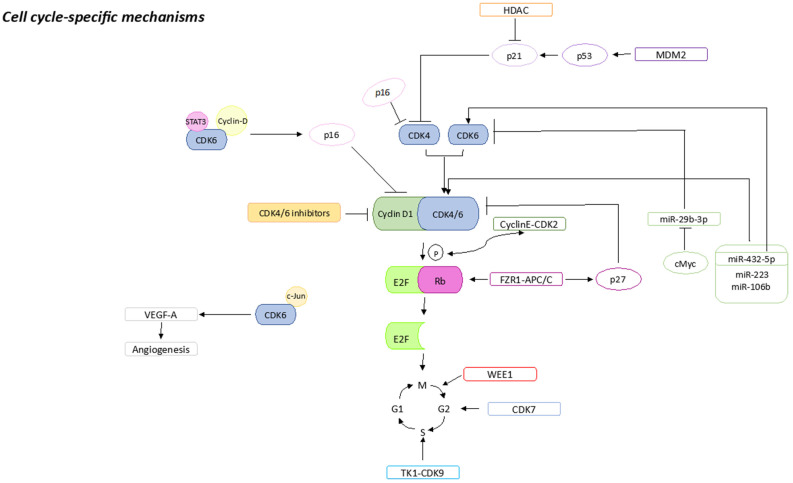
Cell-cycle-specific mechanisms that could be associated with CDK4/6is resistance. Sharp arrows (→) indicate stimulation, blunt arrows (┴) indicate inhibition.

**Figure 2 ijms-24-14427-f002:**
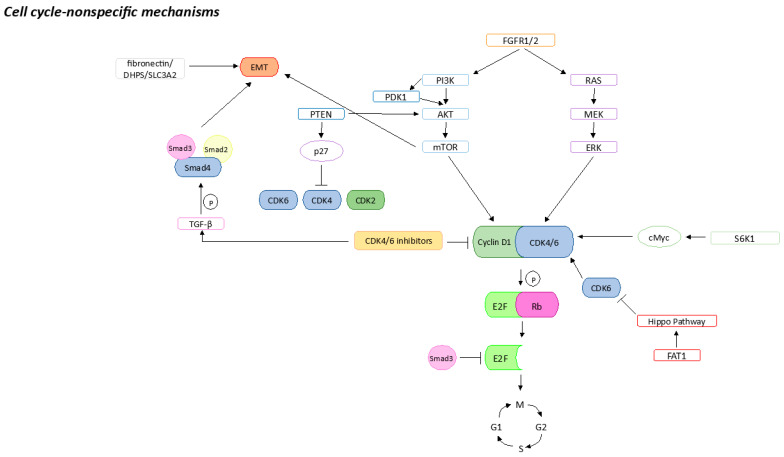
Cell-cycle-non-specific mechanisms that could be associated with CDK4/6is resistance. Sharp arrows (→) indicate stimulation, blunt arrows (┴) indicate inhibition.

## Data Availability

Not applicable.

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
