# Peer review of "Progression after First-Line Cyclin-Dependent Kinase 4/6 Inhibitor Treatment: Analysis of Molecular Mechanisms and Clinical Data"

_ijms, 2023, doi:10.3390/ijms241914427_

Round 1

Reviewer 1 Report

Comments to Villa et al., Progression,,,

This review article describes known machineries whereby breast cancer cells resist to therapy by CDK4/6 inhibitors and therapeutic strategies taken after CDK4/6 inhibition therapy. Regarding the basic/former part, the reviewer saw many similar review articles. Thus, there are few new points in it. But, the latter part seems interesting and unique, as they focused on therapeutic strategies after success or failure of the therapy. Several points are needed to be improved before considering acceptance.  

1.    Many typographic errors like MAINTAIN – MANTAIN. Thanks to ,, is a childish expression. 

2.    Long term and short term issues are mostly overlapped with abovementioned resistant mechanisms. Need to clarify what is the significance of this paragraph. 

3.    In many parts in clinical information, they just gave numbers without providing understandable interpretation. This is a review article. All information should be associated with interpretation. They also need to be kind to readers of basic cancer biology field. 

 Many typographic errors like MAINTAIN – MANTAIN. Thanks to ,, is a childish expression. 

Reviewer 2 Report

This paper Progression after first-line CDK4/6-inhibitor treatment: analysis of molecular mechanisms and clinical data mainly discusses cell cycle-specific and cell cycle non-specific resistance to CDK4/6i with a focus on early and late progression, and the potential therapeutic approaches and

clinical data available: switch to others CDK4/6i, to other single hormonal therapy, to others target therapies (PI3K, mTOR, AKT) and to chemotherapy. The review presents sufficient background investigation with reasonable data analysis, which are in line with the readers interest of International Journal of Molecular Sciences. However, there are still some shortcomings that need to be further improved or explained.

Comments:

Q1. 2.1.1 section, please check Rb or Rb1 is a tumor suppressor protein that plays a pivotal role as a cell cycle checkpoint, as stated in 2.1, Rb means retinoblastoma, not a protein.

Q2. In this paper, CDK4 and CDK6 are always presented as CDK4/6, please explain the differences and relations of these two proteins, and the relevant expressions are suggested to be supplemented in the proper sections.

Q3. In Figure 1, the relationships between CDK7, WEE1 and CDK4/6 are not well reflected, please check.

Q4. As stated in 2.2, Cell cycle-nonspecific mechanisms include the overexpression of factors that are upstream of the cell cycle, such as FGFR and PI3K/AKT/mTOR, resulting in decreased efficacy of CDK4/6i (Figure 2.). Then why are the MDM2, HADC, ect. identified as Cell cycle-specific mechanisms?

Q5. Please check the location of Figure 2.

Q6. The conclusion section should be further improved, as this version only proposes meaningful questions, lacks effective summary of these contents.
